# Properties of Curved Parts Laser Cladding Based on Controlling Spot Size

**Haibo Huang [1],\*, Wenlei Sun [1],\*, Yong Huang [2] and Jiangtong Yu [1]**

[1]   School of Mechanical Engineering, Xinjiang University, Urumqi 830047, China; yujtong@163.com
[2]   Xinjiang Institute of Engineering, Urumqi 830023, China; lishi182@163.com
\*   Correspondence: haibo6669@163.com (H.H.); sunwenxj@163.com (W.S.)

**Abstract:** In this study, a method based on controlling the laser spot size was proposed in the process of curved parts laser cladding, and the coatings obtained by this method were analysed through investigation of the microstructure, microhardness, adhesion property and wear resistance properties. The nonuniform rational B-spline surface (NURBS) reconstruction method was used to obtain the workpiece geometrical characteristics of laser cladding, and through the establishment of a mathematical model, the process of the laser beam working on the curved surface was simplified as the intersection of the cylinder and curvature sphere. Then, the spot size was transformed into the area of a cylinder intersecting with a sphere, and by adjusting the laser head, the size of the laser spot was controlled in the threshold and interpolation points were obtained. The laser cladding trajectory was ensured by these interpolation points, and the experiment was carried out to study the properties of the coating. The results showed that the average coating thickness was about 1.07 mm, and the fluctuation of coating thickness did not exceed 0.05 mm; also, there were no cracks or pores in the layer after penetrant flaw detection. The SEM showed that the grains passed through the transition of plane crystal, cellular crystal, dendrite and equiaxed crystal from the bottom to the top of the layer. After 30 cycles of thermal shock tests, the cladding layer was still well bonded with the substrate and the microhardness and wear resistance were 2 times and 1.4 times higher than that of substrate, respectively.

**Keywords:** laser cladding; curved parts; spot size; microstructure; properties

## 1. Introduction

At present, Fe-based [1–3], Ni-based [4,5], and Co-based [6,7] self-fluxing alloy powders are widely used to repair and strengthen the surface performance in laser cladding. The mechanical properties of coatings are analyzed through microhardness, wear resistance and microstructure properties, and the results shows that coatings can be equal to, or even superior to those of the alloys fabricated by die-casting or forging [8,9]. The repair and recycle industries are therefore greatly enhanced by the advances in laser cladding technology, and have widespread applications in many fields, such as petroleum, chemical, and military [10–12]. Curved parts, such as blades, gears and moulds, are usually key components of equipment and have a high repair and surface strengthening value, but the complex curvature of these parts can lead to variations in the distance or incidence angle of the laser, which affects the final quality of the coating. Many scholars have undertaken relevant research to obtain good cladding layers on complex surfaces. Penaranda et al. [13] proposed an adaptive laser cladding method that considered the geometry of the workpiece, and studied the mathematical relationship between the process parameters and the width of the cladding layer. In the experiment, a changeable cladding width from 0.45 mm to 1.85 mm was obtained by changing the laser power. Liu et al. [14] used the Taguchi method to obtain the optimised laser cladding parameters of a sprocket and then repaired

the damaged sprocket. The results of microstructure analysis indicated that the maximum error was 2.973 mm and that the quality of the cladding layer was good. Zhou et al. [15] found that laser power density would suddenly decrease with the tangential angle of the cladding outline at the bottom of overlapping zone, and pointed out that the width-to-height ratio of the single track was the key factor in pore formation. They also found that a higher power density and lower thermal conductivity could restrain the generation of pores. Zhu et al. [16] studied the influencing rules of the substrate incline angles on the section sizes of cladding layers, and made their robot achieve a variation in angles between 0° and 150°, finally it was used on the hollow parts successfully. Campanelli et al. [17] used the Taguchi method to obtain the optimal laser power, scanning speed, and powder flow rate. They also calculated the optimal overlapping degree by building a mathematical model, and proposed statistical algorithms to evaluate high-density samples.

However, these studies do not consider the variation of spot size during the process of complex parts laser cladding, which is an important process parameter. Therefore, a method of controlling spot size during the laser cladding process is proposed, through the establishment of a mathematical model, in which the spot size is simplified as the area of a cylinder intersecting with a sphere, and then the interpolation points are obtained through the threshold of the spot size. Finally, the method is realized on an ellipsoidal mould and the result is analysed using microstructure, microhardness, adhesion property and wear resistance.

## 2. Establishing the Mathematical Model for Controlling Spot Size

### 2.1. Establishment of the Mathematical Model

For curved parts, the complex surface morphology has an important influence on the scanning spot size, to determine the diameter of laser beam in the cladding process. Wang et al. [18] proposed the equal bow height method, where the laser beam can be simplified as a cylinder. Due to the curved parts having a complex curvature, the laser spot size on the surface is irregular; however, the variable spot size can cause the inhomogeneous distribution of laser energy, which affects the final coating quality. To solve the problem, the surface which is scanned by the laser spot is simplified, as a part of curvature sphere (shown in Figure 1), and the radius of sphere is the reciprocal of the maximum curvature of the point on the surface [19,20]—the problem is therefore transformed into solving the intersection area of the sphere and cylinder. At the beginning of the laser cladding, the laser beam is in the normal direction of the sphere and the axis of the cylinder passes through the centre of the sphere (shown in Figure 2). When the curvature of the workpiece changes, the position and the radius of the curvature sphere changes. In the wake of the moving of the laser beam (cylinder) with its initial attitude, the relative position between the cylinder and the sphere changes; at this time, the second case is formed as follows: the axis of the cylinder does not pass through the centre of the sphere (shown in Figure 3).

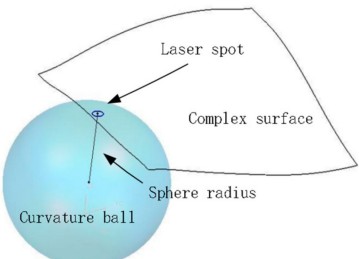

**Figure 1.** The curvature sphere fitting the complex surface.

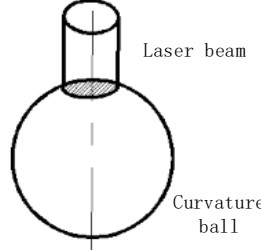

**Figure 2.** The cylinder passing through the sphere centre.

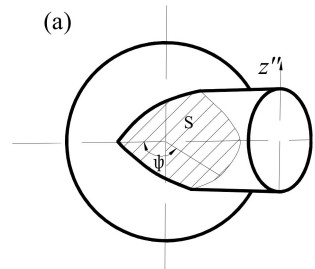

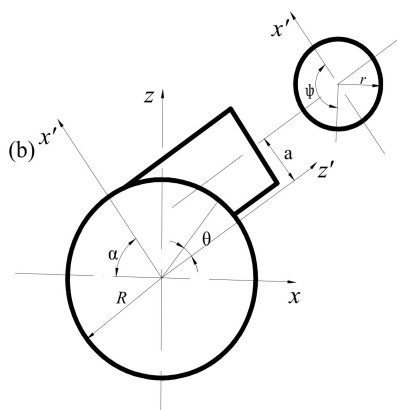

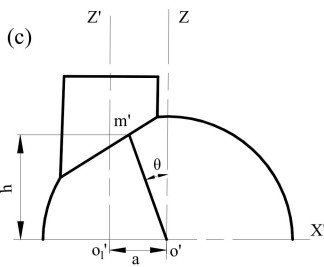

**Figure 3.** The cylinder passing through the sphere without crossing its centre: (**a**) the relative position between cylinder and sphere, (**b**) the side view of the model, (**c**) the model after rotating on the Z axis.

When the axis of the cylinder passes through the centre of the sphere, (as shown in Figure 2), the process of solving the intersection area of the sphere and cylinder is as follows: the radius of the cylinder is set as '$r$', the radius of the curvature ball is '$R$' and the area of the laser beam irradiated on the workpiece is '$A$'. According to mathematical knowledge, the equation of the sphere is $x^2+y^2+z^2=R^2$, and the equation of the cylinder is $x^2+y^2=r^2$. The equation of the irradiated surface is $z=\sqrt{R^2-x^2-y^2}$.

By using the method of finite surface integration, the area is calculated by the Equation (1):

$$A = \iint_{D_{xy}} \sqrt{1 + z_x^2 + z_y^2}\,dxdy = \iint_{D_{xy}} \frac{R}{\sqrt{R^2 - x^2 - y^2}}\,dxdy \tag{1}$$

Solving the equation in the polar coordinates, the result was shown in Equation (2):

$$A = \iint_{D_{xy}} \frac{R}{\sqrt{R^2 - x^2 - y^2}}\,dxdy = R\int_0^{2\pi} d\theta \int_0^r \frac{\rho\,d\rho}{\sqrt{R^2 - \rho^2}} = 2\pi\left(R^2 - R\sqrt{R^2 - r^2}\right) \tag{2}$$

In Figure 3, the axis of the cylinder does not pass through the centre of sphere. With the moving of the laser beam, the cylinder does not pass through the centre of the sphere (the relative position between the two is shown in Figure 3a); '*S*' means the area of the laser beam irradiated on the workpiece. Figure 3b is the side view of the mathematical model, and Figure 3c is obtained by rotating on the Z-axis; with angle '*α*', the parameter '*a*' is the distance from the centre of the sphere to the axis of the cylinder, and the solving process is as follows:

In Figure 3c, the sphere equation is $(x - a)^2 + y^2 + z^2 = R^2$, the cylinder equation is $x^2 + y^2 = r^2$, and the integral equation is shown in Equation (3).

$$S = \iint_{D_{xy}} \frac{R}{\sqrt{R^2 - (x - a)^2 - y^2}}\,dxdy \tag{3}$$

Solving the equation in the polar coordinates,

$$S = \iint_{D_{xy}} \frac{Rs}{\sqrt{R^2 - (x - a)^2 - y^2}}\,dxdy = \int_0^{2\pi} d\theta \int_0^r \frac{Rr}{\sqrt{R^2 - a^2 - r^2 + 2ar\cos\theta}}\,dr$$
$$= \int_0^{2\pi} a\cos\theta\arcsin\frac{r - a\cos\theta}{\sqrt{R^2 - a^2\sin^2\theta}}\,d\theta + \int_0^{2\pi} \sqrt{R^2 - a^2 + 2ar\cos\theta - r^2}\,d\theta \tag{4}$$

## 2.2. The Method of Controlling Spot Size

According to mathematical knowledge, the minimum spot size can be obtained when the laser beam is incident vertically on the surface. Thus, the spot threshold is set based on the minimum spot size. Where the spot size is beyond the threshold, there is an interpolation point, and the orientation of the laser beam is adjusted to be vertical to the surface (as shown in Figure 4).

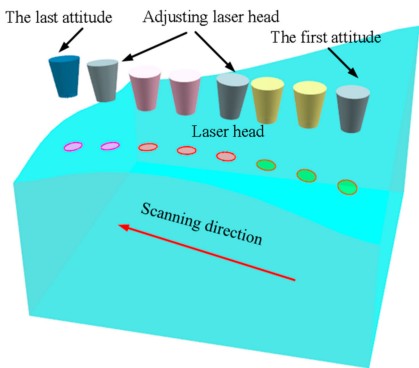

**Figure 4.** Diagram of the adjusting laser head.

## 3. Experimental Procedure

### 3.1. Equipment Used and Reconstruction of the Model

The experiment (Figure 5) was carried out using a fibre laser (manufactured by IPG) with a maximum output power of 4 kW and a six-axis KR30HA KUKA Robot with a powder feeder (DPSF-2); the laser head was equipped with a lateral nozzle. Argon gas was used to deliver powder and prevent melt-pool from being oxidized during the process of laser cladding. Before the experiment, the workpiece was polished with SiC paper (grade 600) and ultrasonicated in acetone for 5 min, then washed in distilled water and dried in the air.

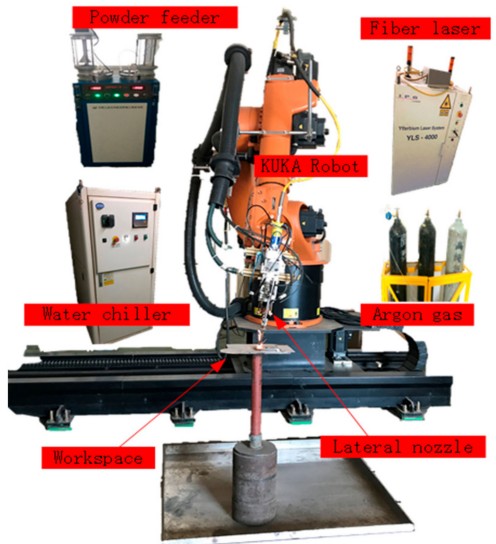

**Figure 5.** The equipment used for laser cladding.

A coordinate measuring machine (CMM) was used to collect the data from the curved surface and the results are shown in Table 1.

**Table 1.** The curved surface data on the workpiece.

| The First Column | | | The Second Column | | | The Third Column | | |
|---|---|---|---|---|---|---|---|---|
| x | y | z | x | y | z | x | y | z |
| 12.00 | −14.71 | 5.00 | 10.00 | −19.66 | 5.00 | 8.00 | −22.93 | 5.74 |
| 12.00 | −14.20 | 5.36 | 10.00 | −19.04 | 5.59 | 8.00 | −22.24 | 6.43 |
| 12.00 | −13.67 | 5.71 | 10.00 | −18.37 | 6.13 | 8.00 | −21.50 | 7.07 |
| 12.00 | −13.13 | 6.03 | 10.00 | −17.68 | 6.64 | 8.00 | −20.71 | 7.66 |
| 12.00 | −7.83 | 8.07 | 10.00 | −11.46 | 9.54 | 8.00 | −10.55 | 11.49 |
| 12.00 | −7.21 | 8.21 | 10.00 | −10.63 | 9.78 | 8.00 | −9.56 | 11.71 |
| 12.00 | −6.59 | 8.35 | 10.00 | −9.80 | 10.01 | 8.00 | −8.57 | 11.91 |

For curved parts, model reconstruction is a very important step before the experiment (the nonuniform rational B-spline (NURBS) surface fitting method [21,22] was used in this experiment). Furthermore, in industrial applications, the accuracy of the double cubic NURBS method can be achieved to $G^2$ level; thereby, the double cubic NURBS fitting method is able to realize the reconstruction of the model. The process of fitting the curved surface [23] was as follows; firstly, the node vector ($u$) was calculated by the accumulative chord length method based on the data in Table 1; then, the node vector $U_i = [0, 0, 0, 0, u_4, \ldots, u_m, 1, 1, 1, 1]$ and its weight $\omega_i$ were calculated, through the parameterisation of the node vector $U_i$, as follows, $x = \frac{u - u_i}{u_{i+1} - u_i}$, $(0 \leq \alpha)$; the control point $d_i (i = 0, 1, \ldots m)$ was obtained based on the matrix operation; then, the cubic nonuniform rational B-spline curve on the $u$ direction

was obtained by using the basic function of the cubic B-spline $N_{i,3}(u)$, taking the control point $d_i(i = 0, 1, \ldots m)$ as the data point on the $v$ direction. Through using the same solving process, as well as the node vector $U_j = [0, 0, 0, 0, u_4, \ldots, u_n, 1, 1, 1, 1]$ and its weight $\omega_j$, the control point $d_j(j = 0, 1, \ldots n)$ and the cubic nonuniform rational B-spline curve on the $v$ direction were obtained. Finally, the double cubic non-uniform rational B-spline surface was obtained based on the Equation (5) (the result is shown in Figure 6) by using MATLAB software.

$$P(u, v) = \frac{\sum_{i=0}^{m} \sum_{j=0}^{n} \omega_{i,j} d_{i,j} N_{i,3}(u) N_{j,3}(v)}{\sum_{i=0}^{m} \sum_{j=0}^{n} \omega_{i,j} N_{i,3}(u) N_{j,3}(v)} \tag{5}$$

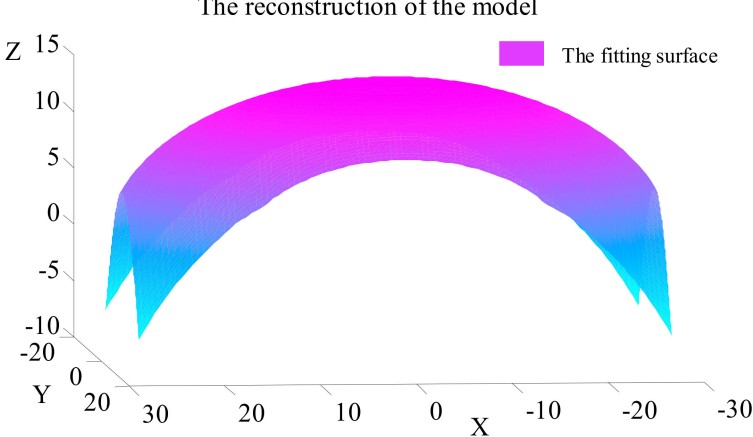

**Figure 6.** The reconstruction of the model.

### 3.2. The Searching Interpolation Points and Experiment Process

In this experiment, the diameter of the spot is 4 mm, and therefore the minimum spot size is 12.56 mm$^2$ when the laser beam is irradiated vertically on a flat plate. To obtain a good quality layer, the allowance error of the spot size cannot exceed 5% of the threshold, that is to say, when the spot size is beyond 13.188 mm$^2$, the orientation of the laser beam is adjusted to be vertical to the surface. Based on this method, the interpolation points of the laser cladding are shown in Figure 7, in which there are 13 scanning tracks and 65 interpolation points.

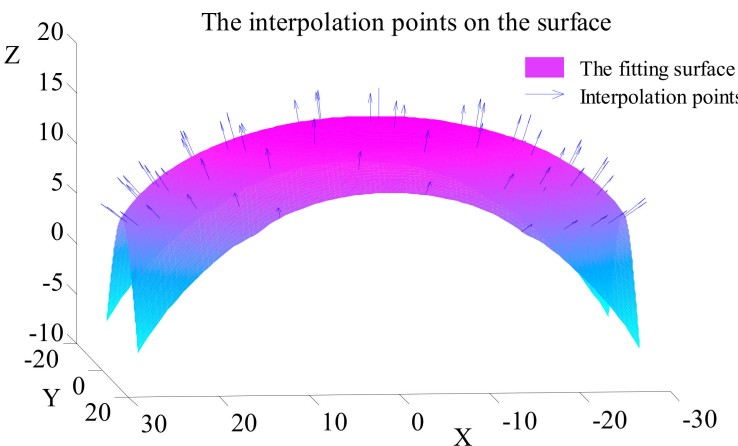

**Figure 7.** The interpolation points on the surface.

The experiment was carried out on an ellipsoid mould, with the substrate material being P20 steel (Figure 8a); its chemical element composition is shown in Table 2. *Fe*-based powder was selected for

the laser cladding, and its composition is shown in Table 3. The parameters of the laser cladding are shown in Table 4. The experiment was carried out based on the obtained interpolation points, and the results after laser cladding are shown in Figure 8b. It was found that the macroscopic morphology of the cladding layer was smooth and dense, and there were no pores or cracks in the layer after penetrant flaw detection (Figure 8c). The coating thickness was measured at multiple locations, and the results are shown in Figure 9. It was found that the thickness existed as different peaks and valleys, but there was little difference between them: the average coating thickness was about 1.07 mm and the maximum thickness fluctuation of the cladding layer did not exceed 0.05 mm.

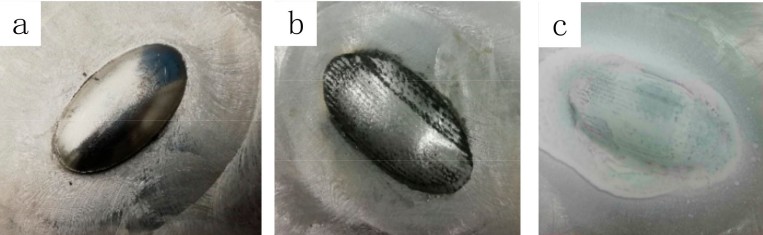

**Figure 8.** The morphology of the mould: (**a**) before laser cladding, (**b**) after laser cladding and (**c**) after penetrant flaw detection.

**Table 2.** The chemical composition of the P20 steel (wt/%).

| Element | C | Cr | Mo | Mn | Si | S | P | Fe |
|---------|-----------|-----------|-----------|-----------|-----------|--------|--------|------|
| Content | 0.28–0.40 | 1.40–2.00 | 0.30–0.55 | 0.60–1.00 | 0.20–0.80 | ≤0.030 | ≤0.030 | Bal. |

**Table 3.** Chemical composition of *Fe*-based alloy powder (wt/%).

| Element | Mn | Cr | Ni | Tb | B | Si | Fe |
|---------|-----|------|-----|-----|-----|-----|------|
| Content | 1.3 | 10.9 | 6.3 | 3.2 | 0.1 | 0.8 | Bal. |

**Table 4.** The laser cladding process parameters of the mould.

| Laser Power/kW | Powder Feeding Rate/(g/s) | Laser Scanning Speed/(mm/s) | Lap Rate/% | Cladding Width/mm | Defocusing Amount/mm |
|----------------|---------------------------|------------------------------|------------|-------------------|----------------------|
| 1.8 | 20 | 4 | 50 | 4 | 16 |

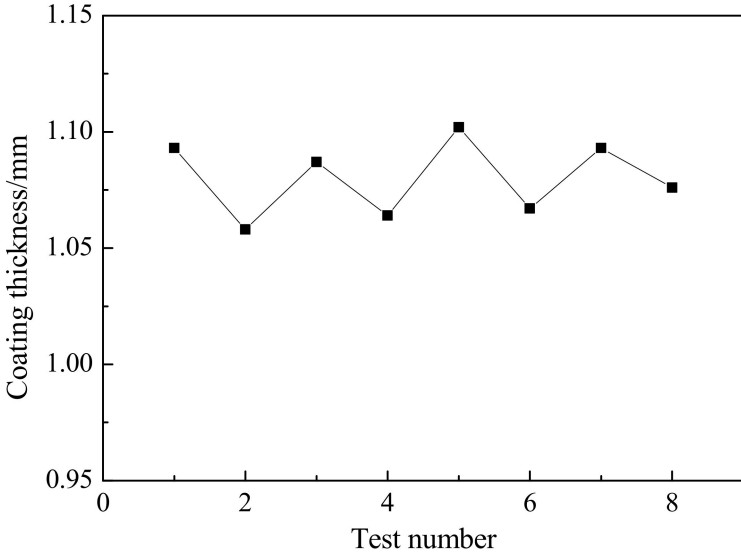

**Figure 9.** Thickness of cladding layer.

## 4. Results and Discussion

### 4.1. Microstructure of Coating

The coatings were cut into standard specimens of 10 mm × 10 mm × 8 mm and analysed with SEM. The etchant in this experiment was aqua regia mixed with HCl and HNO$_3$, with the volume ratio 3:1; the duration of corrosion was 15 s, and the results were shown in Figure 10. The metallographic graph of the overview of the cladding layer was shown in Figure 10a. It was found that a good metallurgical bonding was formed between the substrate and the cladding layer; additionally, there were few pores and cracks, which indicated the rationality of the method. Because the growth of the crystal morphology during solidification mainly depends on the value of *G/R* [9], where '*G*' means the temperature gradient of the liquid phase and '*R*' is the solidification rate of solid phase, the instantaneous value of *G/R* changes with the distance to the solidification interface. The cladding layer is divided into three parts: bottom, middle and top, each having a different micro-morphology [24,25]. Figure 10b shows the bottom of the layer, and the input of laser energy. The surface temperature of the substrate increased rapidly and formed a larger temperature gradient between this layer and the base metal; meanwhile, the solidification rate was the lowest, so the flat interface and directional columnar dendrites were observed in this layer. In the middle of the layer, the solidification rate became higher and the transition region of cellular crystal and dendrite crystal appeared with a micro-segregation of the composition, which is shown in Figure 10c. As for the top of layer, the cooling rate reached the its peak and the microstructure showed an equiaxial grain structure (Figure 10d).

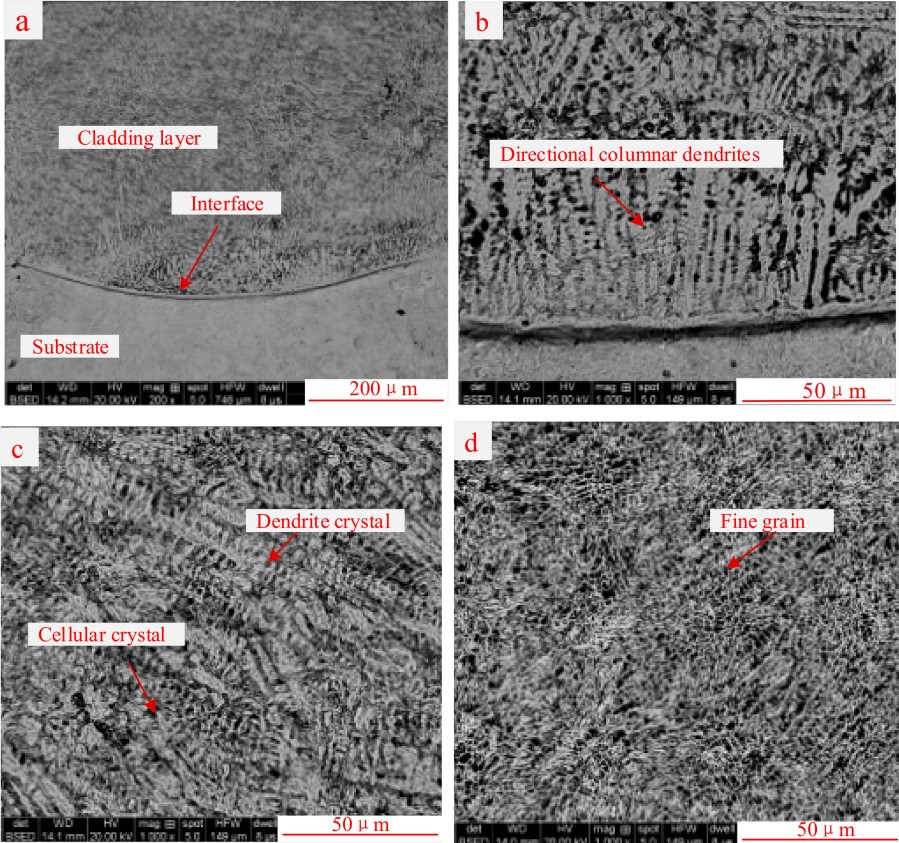

**Figure 10.** SEM micrograph of the coating: (**a**) The overview of the cladding layer, (**b**) the bottom zone, (**c**) the middle zone and (**d**) the top zone.

### 4.2. Microhardness Analysis

A HXD–1000 microhardness tester was used to measure the layer microhardness, with an experimental load of 200 g for 30 s. The test started from the top of the layer to the substrate, with a distance of 0.25 mm between each point; there were three tests in total, in three different positions on the same horizontal line (the results were shown in Figure 11).

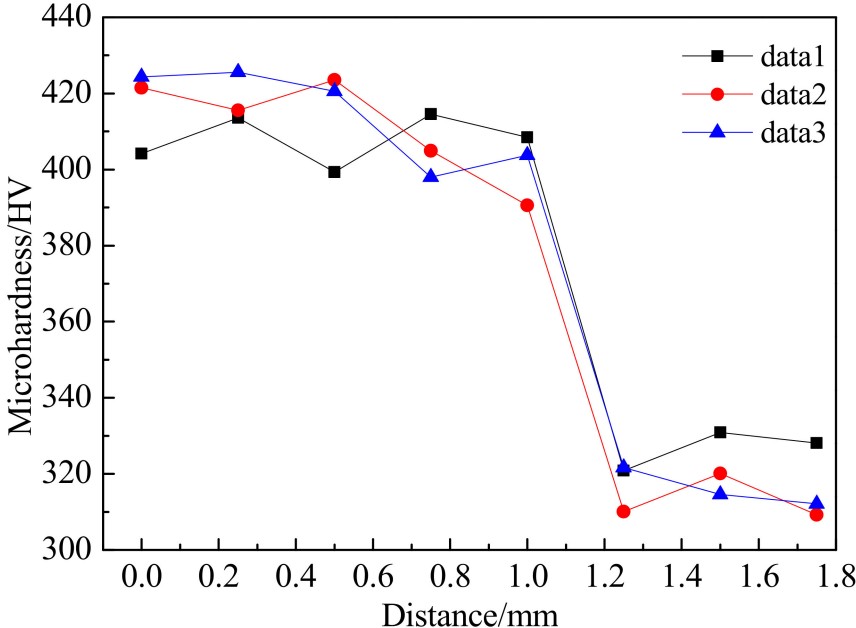

**Figure 11.** Microhardness curves of the layer.

It was found that the microhardness of the layer was about 415 HV, which was 1.3 times higher than that of the substrate. Because the principle of the microhardness measurement is based on the plastic deformation of the material when the load is pressed onto the specimen, the smaller the deformation is, the greater the hardness of the material. The wight content of the element *Cr* in the P20 steel was 1.2%–1.4%, but in the powder material, it was 10.9%. Because the *Cr* element could improve the microhardness in the carbon steel, and the content of the element *Cr* in the layer is higher than that of the substrate, the layer has a higher microhardness.

### 4.3. Thermal Shock Resistance of Cladding Coating

The thermal shock test was used to characterise the adhesion between the layer and substrate in the experiment, using the following process: the size of the sample was 10 mm × 10 mm × 8 mm before the experiment; the workpiece was polished with SiC paper (grade 800); the inside temperature of the muffle furnace was heated to 650°C; the sample was placed in the muffle furnace for 10 min; and the sample was quenched and cooled in water with a temperature of 25°C and dried in the air. The top and the side of the initial cladding layer is shown in Figure 12a,d. After 30 cycles, the result of the layer is shown in Figure 12b,c; the result of the substrate is shown in Figure 12e,f. It was found that both the coating and substrate were oxidized at a high temperature, and some parts had peeled off, but the cladding layer and substrate were still well bonded. The reason for this was mainly that the thermal expansion coefficient of the coating had little difference to that of the substrate, and a good metallurgical bonding between the cladding layer and substrate was formed, creating a layer with a strong thermal shock resistance.

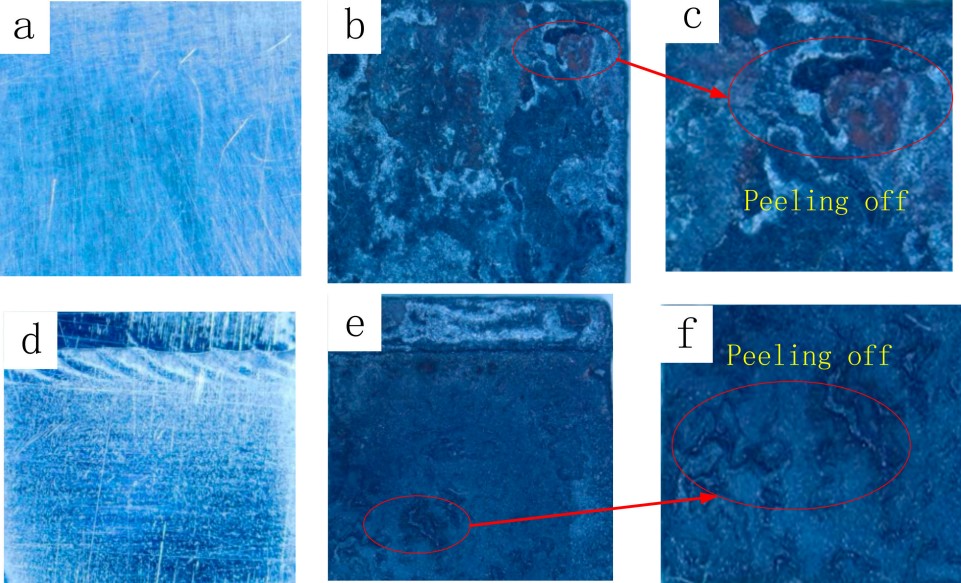

**Figure 12.** The result of thermal shock test (**a**) the top of layer before thermal shock test, (**b**) and (**c**) the top of layer after thermal shock test; (**d**) the side of layer before thermal shock test, (**e**) and (**f**) the side of layer after thermal shock test.

*4.4. Wear Resistance of Cladding Coating*

The wear resistance properties were measured by a pin-on-disc wear testing machine, at room temperature. The coatings were cut into standard specimens of 30 mm × 7 mm × 6 mm; next, the surface of the coating was polished with SiC paper (grade 1200) and the sample was fixed on the wear testing machine. The countermaterial was the GCr15 steel after quenching with the normal load of 200 N; the diameter and thickness of the disc friction pair were, respectively, Φ40 mm and 10 mm; the surface roughness of the friction pair was 6.3 μm; and the sliding velocity was 200 r/min with an unidirectional way. The loss of the mass was measured every 40 min, and the results were shown in Figure 13. After 160 min, the wear mass of the coating was 21.08 mg, which was less than the 9.94 mg of the substrate. The main reason for the difference was that the surface grain structure of the layer was refined (as shown in Figure 14); the size of the surface grain was in the range of 2 μm–4 μm, and there were many grain boundaries that could resist plastic deformation. In addition, the surface hardness was improved after laser cladding, and a high level of hardness improved the wear resistance of materials; therefore, the resistance property of the layer was better than the P20 steel.

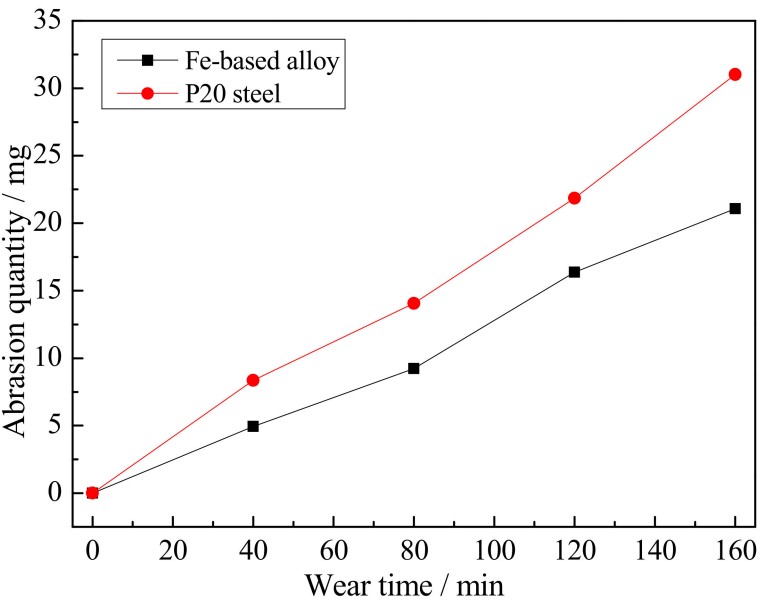

**Figure 13.** The loss of mass after the wear experiment.

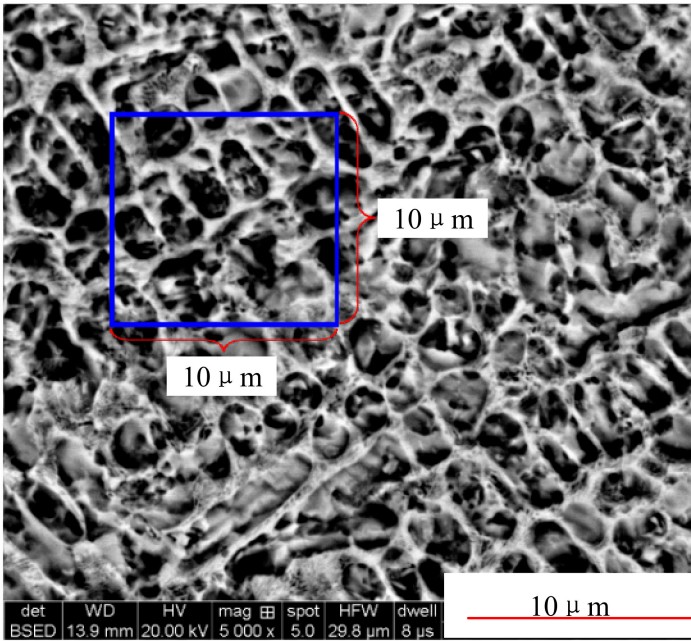

**Figure 14.** Grain structure of the upper surface.

## 5. Conclusions

In this paper, a method based on controlling the laser spot size was used in the process of curved parts laser cladding, and the coatings obtained were analysed through investigation of the microstructure, microhardness, adhesion property and wear resistance properties. Our conclusions are shown below.

(1)　The smooth and dense surface coatings were obtained by using the method, and there were no pores or cracks after the penetrant flaw detection;

(2)　Through the analysis of SEM, good metallurgical bonding between the substrate and layer was obtained, with the presence of plane crystal, cellular crystal, dendrite and equiaxed crystal in the coatings;

(3)     Compared with the substrate, the wear mass of coating was 9.94 mg less than that of substrate after 160 min, and the adhesion of the coating and base was still well-combined after 30 thermal shock tests. In addition, the microhardness was improved 1.3 times more than that of substrate, which showed the rationality and feasibility of the method.

**Author Contributions:** Funding acquisition, W.S.; Validation, Y.H. and J.Y.; Writing—original draft, H.H. All authors have read and agreed to the published version of the manuscript.

**Funding:** The author would like to appreciate the support from the Karamay major project, China (2018ZD002B).

**Acknowledgments:** Thanks to my tutor and all my colleagues in the School of Mechanical Engineering, Xinjiang University. And thanks eceshi(www.eceshi.cn) for the SEM analysis.

**Conflicts of Interest:** The authors declare no conflict of interest.

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
