# Peer review of "Properties of Curved Parts Laser Cladding Based on Controlling Spot Size"

_applsci, doi:10.3390/app10020728_

Round 1

Reviewer 1 Report

In this paper entitled ”Properties of curved parts laser cladding based on controlling spot size” the authors have studied a method based on controlling laser spot size in the process of curved parts laser cladding. The topic could be of interest for those using laser technology but there are many points that should be addressed and therefore a major revision should be done.

Comments:

Abstract: Please, remove acronym NURBS from abstract and replace by the full description. Line 72: The problem is solved in two situations but there is no description about why is that so. Is it because for the first one, when the cylinder passes through the sphere center it is symmetric while when it does not pass through the center it is not symmetric?. Line 78: For equations 1 and 2 the parameter ‘r’ (means the radius of the cylinder) is not shown. Line 89: Why the experiments were carried out using fiber lases (IPG)?. Line 99: Please, explain what does G2 level stands for. Please, explain better the connection between equation (3) and Fig. 5. Why Fig. 5 shows X and Z parameters while equation (3) does not contain these parameters?. Why the experiment was carried out in an ellipsoid mold while for Fig. 1 it is a sphere and why P20 steel?. Please, make Fig. 9 larger and explain why the solidification rates are different depending on the location in the layer. Please, make Fig. 11 larger. It is difficult to see the peeling off. Line 182. “The surface grain structure of the layer was refined”. Please give numbers of grain size to be able to assess how much refined the microstructure is compared with other parts of the coating. I do not see where in the paper the spot size is linked to the microstructure and hardness.

Reviewer 2 Report

This is an interesting work that fits within the scope of this Journal. However, prior to publication several points need to be addressed. My comments/suggestions are given hereafter:

1. In the abstract you mention the fluctuation of the thickness of the coating, but you do not mention its thickness.

2. What is the size of Fe-based powders?

3. Did you perform any surface preparation/treatment before the cladding process? If yes, then you should mention this in the experimental part.

4. After cladding, did you perform any surface finishing process? Roughness can have a strong effect on friction and wear measurements.

5. In the experimental part, there is no description of hardness, adhesion and tribological tests. A detailed description should be added in the manuscript. For example, when performing wear tests: what was the countermaterial (composition, geometry, dimensions, roughness etc.), what was the sliding velocity and motion (reciprocating or unidirectional)?

6. How many tests did you perform per condition?

7. Which etchant did you use for the metallographic observation of the coating?

8. In figure 10, when you refer to data 1-3, do you mean repeats?

9. You conclude that the higher hardness is due to the finer microstructure of the coating. However, you should also provide an image of the substrate for comparison.

10. Legend of figure 11 does not provide any information.

11. Why did you select such a high load 900 N? If you used a point contact (ball-on-flat), this would result in extremely high contact pressures (overstressing coating?).

12. You conclude that the coating has improved wear properties because it can resist the abrasion particles. However, the wear mechanism was not confirmed by presenting an image of the wear track. Typically, in a dry steel vs steel contact, adhesive wear also takes place.

Round 2

Reviewer 1 Report

The authors have properly addressed all the queries and therefore the paper is ready for publication.

Reviewer 2 Report

Thank you for addressing my comments/suggestions. From my side this work is now appropriate for publication.